# Pathogenic *Leptospira* Infections in Hubei Province, Central China

**DOI:** 10.3390/microorganisms11010099

**Published:** 2022-12-30

**Authors:** Jiale Xu, Jintao Chen, Chaorui Xiong, Lingxin Qin, Bing Hu, Manqing Liu, Yuting Ren, Yirong Li, Kun Cai, Liangjun Chen, Wei Hou

**Affiliations:** 1State Key Laboratory of Virology/Department of Laboratory Medicine/Hubei Provincial Key Laboratory of Allergy and Immunology, School of Basic Medical Sciences, Zhongnan Hospital, Wuhan University, 185 Donghu Road, Wuhan 430071, China; 2Institute of Health Inspection and Testing, Hubei Provincial Center for Disease Control & Prevention, 6 Zuodaoquan Road, Wuhan 430079, China; 3Division of Virology, Wuhan Center for Disease Control & Prevention, 288 Machang Road, Wuhan 430015, China; 4School of Public Health, Wuhan University, 185 Donghu Road, Wuhan 430071, China

**Keywords:** pathogenic *Leptospira*, wild rodent, zoonosis, central China

## Abstract

Leptospirosis is an important zoonosis that is caused by pathogenic *Leptospira*, which is considered to be a re-emerging infectious disease in many countries. Rodents are the most important reservoirs for both human and animal infection. An epidemiological survey of pathogenic *Leptospira* in rodents is important for the prevention and control of leptospirosis. In this study, a total of 964 rodents were captured from six cities in Hubei Province, and two pathogenic *Leptospira* species (*L. interrogans* and *L. borgpetersenii*) were detected using nested PCR with an overall prevalence of 4.8%. *L. interrogans* was distributed in five sampling sites, which may be the dominant species of pathogenic *Leptospira* in Hubei Province. In addition, *Rattus norvegicus* showed a relatively high infection rate, which may play an important role in the transmission and infection of pathogenic *Leptospira*. This study reveals the prevalence of pathogenic *Leptospira* in wild rodents in Hubei Province, suggesting that the risk of leptospirosis infection in Hubei Province still exists.

## 1. Introduction

Leptospirosis caused by the pathogenic *Leptospira* spp. is one of the most significant zoonotic diseases, mainly distributed in tropical and subtropical countries [1,2]. It is estimated that there are 1.0 million cases and 58,900 deaths due to leptospirosis worldwide every year [3]. The genus *Leptospira* can be classified into pathogenic, saprophytes and intermediate species according to their virulence status [2,4]. Among the pathogenic group, *L. interrogans*, *L. borgpetersenii* and *L. kirschneri* are the main pathogenic bacteria responsible for human leptospirosis worldwide [5,6,7].

*Leptospira* is excreted from the kidneys of host animals into urine and then contaminates soils and rivers. Human infection may result from direct contact with the urine of infected animal hosts or the ingestion of contaminated water [8,9]. At least 200 species of animals have been confirmed as natural carriers of pathogenic *Leptospira*, especially rodents, which are considered the main reservoir for *Leptospira* infection in humans [9,10,11,12]. Therefore, the surveillance and investigation of pathogenic *Leptospira* in wild rodents will contribute to understanding animal-to-human transmission and tracking for leptospirosis.

Leptospirosis, a category B notifiable infectious disease in China, was first reported in 1934 and has been a mandatory notifiable disease since 1955 [13]. To date, more than 2.5 million cases and over 20,000 deaths have been reported in China [14]. Hubei Province is known as the “land of a thousand lakes”, and most areas possess a subtropical humid monsoon climate [15,16]. Abundant rainfall, rice cultivation and high density of rodents provide a favorable environment for the survival and transmission of *Leptospira.* According to incidence of leptospirosis, China can be divided into four regions [12]. Hubei belongs to region B (the temperate region located in the middle and lower areas of the Yangtze River) with the highest human leptospirosis incidence [12]. Over the past few decades, China has dramatically reduced the incidence of human leptospirosis through a series of measures [17]. However, some studies have shown that there are persistent high risk leptospirosis clusters along the Yangtze River basin, suggesting that the risk of leptospirosis infection in Hubei Province should not be ignored [18]. According to the data available from the official departments in mainland China, the total number of confirmed leptospirosis cases was 403 in 2021 in China (http://www.nhc.gov.cn/jkj/s3578/202204/4fd88a291d914abf8f7a91f6333567e1.shtml, accessed on 5 November 2022), while from 2007 to 2018, the total incidence in Hubei Province was 0.4022 cases per 100,000 people, which was at a high level [19]. Therefore, it is of great significance to monitor and investigate the prevalence of pathogenic *Leptospira* in rodents in Hubei Province for the prevention and control of human leptospirosis.

## 2. Materials and Methods

### 2.1. Rodent Collection and Sample Processing

Rodents were captured with cage traps loaded with food bait in Wuhan, Shiyan, Huangshi, Jingzhou, Xiangyang and Xianning cities in Hubei Province in 2021 (Figure 1). The topography of Hubei is generally high in the west and low in the east. Shiyan is located in the northwestern Hubei mountain region with an average altitude of 736.9 m, whereas Xiangyang, Jingzhou, Wuhan, Xianning and Huangshi are in the hilly plain area, with an average altitude of 347.6, 42.7, 37.1, 189.6 and 109.4 m, respectively. Rodent species were identified using morphological examination and further confirmed using mitochondrial *cyt*-b gene sequence analysis. All rodents were captured alive and euthanized based on humanitarian principles. The kidney tissue samples were collected and stored at −80 °C until further use.

### 2.2. Screening and Molecular Characterization of Pathogenic Leptospira

Total DNA was extracted from kidney tissue samples according to the instructions of the DNA extraction kit (Omega, Doraville, CA, USA). All samples were screened for the presence of *Leptospira* using nested PCR, and the primers targeting the conserve region of 16S rRNA (*rrs*) gene in pathogenic *Leptospira* species were selected [20]. Furthermore, the partial *LipL32* and *secY* genes, which are widely used in phylogenetic analyses [21,22,23], were also recovered from the positive samples to better identify and characterize different species in positive samples. The PCR products with expected size were sent for sequencing (Wuhan Gene Create Biological Engineering Co., Ltd., China). To prevent contamination, dedicated pipets and filtered tips were used, and each operation, including PCR mixture preparation, template addition and agarose gel electrophoresis were performed in separate rooms. The primer sequences used in this study are listed in Appendix A.

### 2.3. Genetic and Phylogenetic Analysis

All the nucleotide sequences were assembled and edited using SeqMan program (DNASTAR, Madison, WI, USA) and then compared with reference sequences using basic local alignment search tool (BLAST, https://blast.ncbi.nlm.nih.gov, accessed on 15 July 2022). The sequences obtained in this study and the reference sequences downloaded from GenBank were aligned and calibrated using Clustal W in MEGA7. The nucleotide (nt) sequence identities were calculated using the MegAlign program available within the Lasergene software package (version 7.1). The maximum likelihood method (ML) and general time reversible (GTR) model was used to reconstruct the phylogenetic trees with bootstrap support values calculated from 1000 replicates implemented in MEGA7. All the sequences obtained in this study have been submitted to GenBank under the accession numbers OP860838-OP860883 (*rrs*), OP874962-OP875007 (*secY*) and OP875008-OP875053 (*LipL32*).

### 2.4. Statistical Analysis

The statistical analysis was performed using the Statistical Package for Social Sciences Version 27.0 software (SPSS, Chicago, IL, USA). Fisher exact test was performed to determine the differences in *Leptospira* positivity rates between different collected locations and rodent species [24,25], and significance was defined at a *p* value of 0.05. The 95% confidence interval was calculated using the Epitools (https://epitools.ausvet.com.au/trueprevalence, accessed on 26 October 2022).

## 3. Results

### 3.1. Detection of Pathogenic Leptospira

A total of 964 rodents belonging to 6 different species were captured from six cities in Hubei Province in 2021 (Figure 1), including 389 *Rattus norvegicus*, 192 *Apodemus agrarius*, 107 *Mus musculus*, 270 *Rattus tanezumi*, 5 *Niviventer confucianus* and 1 *Micromys minutus* (Table 1). Nested PCR targeting a conserve region of 16S rRNA (*rrs*) gene was performed to detect pathogenic *Leptospira.* We found 46 *Leptospira*-positive rodents, including 40 *Rattus norvegicus* and 6 *Apodemus agrarius*, with an overall positive rate of 4.8% (95% confidence interval: 3.6–6.3%).

*R*. *norvegicus* showed a significantly highest infection rate of 10.3% among all of the rodent species. Geographically, *Leptospira* detected in this study showed variable prevalence in the six cities, ranging from 0 to 20%. Fisher’s exact test revealed highly significant differences in the distribution of leptospiral prevalence across the rodent species and collected locations among these pathogenic *Leptospira* (*p* < 0.001). In addition, *Leptospira* was detected in *R*. *norvegicus* in five of the six sampling areas, with positive rates ranging from 5.7 to 25.4%, and the infection rates of *R*. *norvegicus* collected in Xianning city were the highest.

### 3.2. Molecular Characterization of Pathogenic Leptospira

The *LipL32* and *secY* genes of pathogenic *Leptospira* were amplified from all positive samples to better characterize the species of the detected pathogenic *Leptospira* strains. BLAST analysis of the *rrs* gene sequences showed that the *Leptospira* detected in our study were identified as *L. interrogans* (*n* = 44) and *L. borgpetersenii* (*n* = 2).

The *rrs*, *LipL32* and *secY* gene sequences of *L. interrogans* obtained in this study shared 99.7−100%, 97.6−100% and 98.7−100% nucleotide homology with each other, respectively. In addition, the nucleotide homology with the corresponding gene sequences of *L. interrogans* retrieved from GenBank was 99.4−100%, 98.3−100% and 98.7−100%. Meanwhile, the *rrs*, *LipL32* and *secY* gene sequences of the two *L. borgpetersenii* obtained shared 100% nucleotide homology with each other, respectively, and showed 99.6−100%, 98.6−99.1% and 97.6−99.7% nucleotide homology with other known gene sequences.

The ML phylogenetic tree constructed based on the *rrs* (1057 bp), *LipL32* (587 bp) and *secY* (833 bp) gene sequences showed a similar topology; all the 46 *Leptospira* strains detected in this study belonged to the pathogenic group and clustered into two clades, *L. interrogans* and *L. borgpetersenii* (Figure 2).

### 3.3. Geography and Host Analysis of Pathogenic Leptospira

The Fisher’s exact test revealed significant differences in the distribution of leptospiral species prevalence across the collected sites (*p* = 0.032) and rodent species (*p* = 0.014) among the 46 strains (Table 2). *L. interrogans* was widely distributed in Wuhan, Xianning, Jingzhou, Xiangyang and Huangshi. Moreover, *L. interrogans* was detected in both *R. norvegicus* and *A. agrarius*, whereas *L. borgpetersenii* was found only in *A. agrarius*.

In this study, *R. norvegicus* showed a high positive rate in the captured rodents, and *L. interrogans* showed a high infection rate and a wider prevalence among the two *Leptospira* species, indicating that *R. norvegicus* may be the main carrier of pathogenic *Leptospira,* and *L. interrogans* was the dominant species, followed by *L. borgpetersenii*.

## 4. Discussion

Leptospirosis is an important but neglected zoonotic disease with insufficient research attention in relation to burden, and the incidence rates are significantly underestimated due to a lack of epidemiological work, insufficiently rapid diagnostics and misdiagnosis [26,27]. It has caused serious public health problems worldwide, especially in East and Southeast Asian countries such as China, South Korea and Vietnam [12,28,29]. In recent decades, China has effectively reduced the incidence and mortality of leptospirosis nationwide which is attributed to public education, vaccination, rural environmental improvements and better sanitation [30,31]. Although the incidence of leptospirosis has significantly decreased, local outbreaks are still frequently reported in certain areas [31,32]. Rodents are considered to be the main carriers of *Leptospira* and are also an important contributor to leptospirosis in humans [33,34]. Hubei Province is one of the main endemic areas of leptospirosis [12]; however, studies on the epidemiology of pathogenic *Leptospira* in rodents in Hubei Province are very limited.

In this study, 46 pathogenic *Leptospira* strains were detected, including 44 *L. interrogans* and 2 *L. borgpetersenii* with positive rates of 4.6 and 0.2%, respectively. It is worth noting that these two *Leptospira* strains are the main causes of leptospirosis in China, of which *L. interrogans* has caused at least 60% of the human cases of leptospirosis in China historically [12,14]. The high prevalence of *L. interrogans* has also been reported in previous studies [20,24,25,35]. All these suggest that *L. interrogans* may be the main pathogenic *Leptospira* circulating in nature.

Studies have shown that the prevalence of *Leptospira* varies in different geographical ecological environments [24,36]. In our study, the prevalence rate of rodents carrying *Leptospira* is high in hilly and plain areas, such as Wuhan, Jingzhou, Xianning, etc., whereas no rodent was found to be infected with *Leptospira* in Shiyan, located in a mountain area. This observation is consistent with a previous study, which found that all *Leptospira*-positive rodents were detected in low-altitude locations, whereas none of the rodents in higher locations hosted *Leptospira* [23]. Rodents may migrate between habitats, and these movements may involve rodents infected with *Leptospira* [36]. The differences in topography and altitude in these areas in this study may affect this migration and thus the distribution of *Leptospira*.

Due to their abundance and close association with human habitats, rodents play an important role in the transmission of *Leptospira*. Among them, *Rattus norvegicus* is the main source of *Leptospira* infection in humans [10,37,38]. Previous studies have reported that *R. norvegicus* is the main host of *L. interrogans* in urban areas [39]. It was also found that *R. norvegicus* accounted for 38.1% of *Leptospira* infected animals, although it accounted for only 15.6% of the captured animals [40]. In our study, most of the *Leptospira* detected in rodents captured in Hubei Province were also from *R. norvegicus*. All these indicated that *R. norvegicus* was the main species involved in *Leptospira* transmission.

It is widely accepted that the presence of rivers near human settlements increases the infection risk of *Leptospoira*, and flooding has also been linked to outbreaks of leptospirosis [41,42,43,44]. In this study, we sampled rodents in Hubei Province and found the prevalence of two pathogenic *Leptospira* species in wild rodents, with *L. interrogans* as the dominant species and *R. norvegicus* as the main host. The prevalence of *Leptospira* varied in Wuhan, Huangshi, Jingzhou, Xiangyang and Xianning, which may be related to the region, representativeness and density of the samples we captured. Although based on limited rodent samples, these results reveal that genetically diversified pathogenic *Leptospira* are disseminating among wild rodents in Hubei Province and indicate a potential risk of rodent-derived leptospirosis in Hubei Province.

## Figures and Tables

**Figure 1 microorganisms-11-00099-f001:**
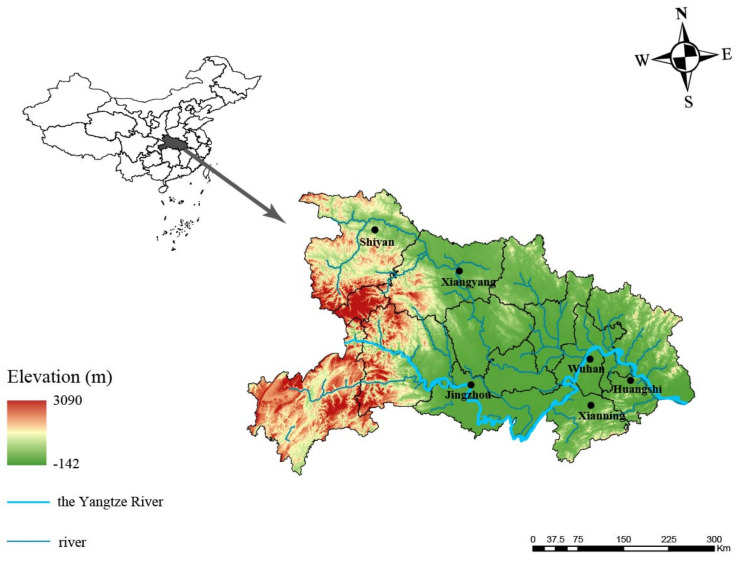
Geographic map showing the location of sampling sites. Samples collected from six cities of Hubei Province are shown using circles. The map was created in ArcGIS 10.8 software (ESRI Inc., Redlands, CA, USA) and Adobe illustrator, Version CC2018 (Adobe, San Jose, CA, USA). The black dots indicate the sampling regions in this study.

**Figure 2 microorganisms-11-00099-f002:**
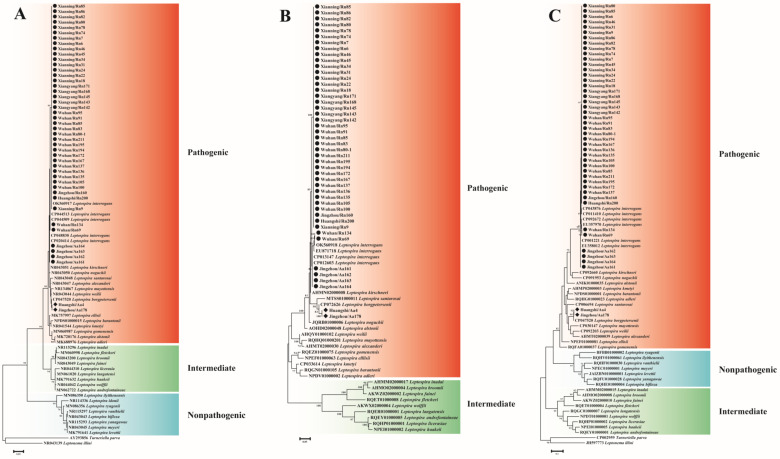
Maximum-likelihood phylogenetic trees based on the nucleotide sequences of *rrs* (**A**), *LipL32* (**B**) and *secY* (**C**) gene of pathogenic *Leptospira*. Sequences belonging to this study are displayed by black circles and diamonds. *L. interrogans* and *L. borgpetersenii* were identified among these 46 *Leptospira* strains. Bootstraps values > 70 are indicated at appropriate nodes. (Aa: *Apodemus agrarius*, Rn: *Rattus norvegicus*).

**Table 1 microorganisms-11-00099-t001:** Prevalence of pathogenic *Leptospira* according to locality and rodent species in Hubei.

Species	Jingzhou (%)	Xiangyang (%)	Shiyan (%)	Huangshi (%)	Xianning (%)	Wuhan (%)	Total (%)
*Rattus norvegicus*	1/8 (12.5)	5/87 (5.7)	0/70 (0)	1/11 (9.1)	16/63 (25.4)	17/150 (11.3)	40/389 (10.3)
*Apodemus agrarius*	5/125 (4.0)	0/45 (0)	0/3 (0)	1/19 (5.3)	0 (0)	0 (0)	6/192 (3.1)
*Mus musculus*	0/23 (0)	0/13 (0)	0/69 (0)	0/2 (0)	0 (0)	0 (0)	0/107 (0)
*Rattus flavipectus*	0/46 (0)	0/55 (0)	0/64 (0)	0/21 (0)	0/17 (0)	0/67 (0)	0/270 (0)
*Niviventer confucianus*	0 (0)	0 (0)	0/4 (0)	0/1 (0)	0 (0)	0 (0)	0/5 (0)
*Micromys minutus*	0/1 (0)	0 (0)	0 (0)	0 (0)	0 (0)	0 (0)	0/1 (0)
Total (%)	6/203 (3.0)	5/200 (2.5)	0/210 (0)	2/54 (3.7)	16/80 (20.0)	17/217 (7.8)	46/964 (4.8)

**Table 2 microorganisms-11-00099-t002:** Species distribution of the 46 pathogenic *Leptospira* among the different rodents and cities in Hubei.

City	No. of Individuals	No. Per Species (Prevalence, %)	
		*L. interrogans*	*L. borgpetersenii*
Jingzhou	203	5 (2.5)	1 (0.5)
Xiangyang	200	5 (2.5)	0
Shiyan	210	0	0
Huangshi	54	1 (1.9)	1 (1.9)
Xianning	80	16 (20)	0
Wuhan	217	17 (7.8)	0
Species			
*Rattus norvegicus*	389	40 (10.3)	0
*Apodemus agrarius*	192	4 (2.1)	2 (1.0)
*Mus musculus*	107	0	0
*Rattus flavipectus*	270	0	0
*Niviventer confucianus*	5	0	0
*Micromys minutus*	1	0	0
Total (%)	964	44 (4.6)	2 (0.2)

## Data Availability

The sequences obtained in this study have been submitted to GenBank under the accession numbers OP860838-OP860883 (*rrs*), OP874962-OP875007 (*secY*) and OP875008-OP875053 (*LipL32*).

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
