# Peer review of "Pathogenic Leptospira Infections in Hubei Province, Central China"

_microorganisms, 2022, doi:10.3390/microorganisms11010099_

Round 1
Reviewer 1 Report
Manuscript ID. microorganisms-2114793
Title: Pathogenic Leptospira infections in wild rodents from Hubei Province….
Authors: Jiale Xu, et al.
The author(s) mainly described in this study the importance of Leptospirosis as a zoonosis that caused by pathogenic Leptospira, which is a re-emerging infectious disease in many countries. Rodents are the most important reservoirs for human infection. The authors collected a total of 964 rodents from six cities in Hubei Province to perform epidemiological survey of pathogenic Leptospira derived from rodents. Two pathogenic Leptospira species (L. interrogans and L. borgpetersenii) were detected by nested PCR. In addition, Rattus norvegicus showed a relatively high infection rate. According to these data, the prevalence of pathogenic Leptospira in wild rodents in Hubei Province showed a unique aspect and existence of the leptospirosis risk.
Overall, this manuscript is reasonably well described. However, several concerning points need to be addressed.
1. The authors maintain that Hubei Province is known as the "land of a thousand lakes", and most areas possess the subtropical humid monsoon, monsoon climate, abundant rainfall, and rice cultivation as the reason for that R. norvegicus is the main species involved in Leptospira transmission. Then the authors should explain the reason for the difference in isolation rates of Leptospira for Wuhan, Huangshi, Jingzhou, Xiangyang, and Xianning locations except Shiyan, in Hubei Province.
2. In introduction, the authors represent “China can be divided into four regions”, however reference [12] shows “three regions”.
3. The authors should indicate the Yangtze River in Figure 1.
4. Where is region B? The authors should explain it.
5. In results and Table 1, the authors state “…and the infection rates of R. norvegicus collected in Xianning city was the highest.”, however it is not sure in Table 1. The authors should add isolation rates to each column data.
6. The authors should make a ruled line between “Wuhan” and ”Species”.
7. In Table S1, amplicon size of rrs, LipL32, and secY gene are showed 1100bp, 730bp and 1200bp, respectively, however the ML phylogenetic tree constructs based on the 1057bp, 587bp, and 833bp gene sequences. Why are there fewer nucleotide reads (approximately 69%) in only secY gene?
Author Response
We appreciate your careful review for our manuscript. Please see the attachment.

Reviewer 2 Report
Title: …wild Rodents in Hubei Province.
I think “in Hubei Province” is better.
Page 3 in 2.2.
How did you read the genes? What equipment did you use? Or did you entrust it to some company?
Page 6 in 4. Discussion
Leptospirosis is an important but neglected zoonotic disease [19].
19. Bharti, A.R.; Nally, J.E.; Ricaldi, J.N.; Matthias, M.A.; Diaz, M.M.; Lovett, M.A.; Levett, P.N.; Gilman, R.H.; Willig, M.R.; Gotuzzo, E.; et al. Leptospirosis: a zoonotic disease of global importance. Lancet Infect Dis 2003, 3, 757-771.
I think this reference paper is a top-class paper but very old, as 2003. And please write in detail the meaning of “neglected zoonotic disease”
Page 6 in 4. Discussion
In our study, the prevalence rate of rodents carrying Leptospira is high in hilly and plain areas, such as Wuhan, Jingzhou, Xianning, etc., while no rodent was found to be infected with Leptospira in Shiyan located in high mountain area. This observation is consistent with previous study ,which found that all Leptospira-positive rodents were detected in low-altitude locations while none of the rodents in higher locations hosted the Leptospira [32].
You wrote these sentences in Discussion. But there is no information about high mountain area (Shiyan), low-altutude area (Wuhan, Jingzhou, Xianning) in 2.1 Rodent collection and sample processing, Materials and methods. In addition, I cannot understand Shiyan located in high mountain area in Figure 1. Please write information about altitude of sampling regions. And please Figure 1 rewrite, because we dont understand "Shiyan is high mountain area".
In discussion.
Difference of Leptospira prevalence is only high and low altitude? How about temperature?
Page 6 in 4. Discussion
Hubei Province is known as the "land of a thousand lakes", and most areas possess the subtropical humid monsoon and monsoon climate [42,43]. Abundant rainfall, rice cultivation and high density of rodents provide a favorable environment for the survival and transmission of Leptospira.
You wrote upper sentences in Discussion. I think you should write these informations in Introduction and/or Materials and methods.
Author Response

(The authors gave the same response as above.)
